# More’s the Same—Multiple Hosts Do Not Select for Broader Host Range Phages

**DOI:** 10.3390/v15020518

**Published:** 2023-02-13

**Authors:** Jupiter Myers, Joshua Davis II, Megan Lollo, Gabriella Hudec, Paul Hyman

**Affiliations:** 1Department of Biology and Toxicology, Ashland University, Ashland, OH 44805, USA; 2Blood Bank, Akron City Hospital, Akron, OH 44304, USA; 3Cleveland Cord Blood Center, Cleveland, OH 44128, USA

**Keywords:** bacteriophage, host range, phage isolation, host-parasite interaction, phage therapy

## Abstract

Bacteriophage host range is a result of the interactions between phages and their hosts. For phage therapy, phages with a broader host range are desired so that a phage can infect and kill the broadest range of pathogen strains or related species possible. A common, but not well-tested, belief is that using multiple hosts during the phage isolation will make the isolation of broader host range phage more likely. Using a *Bacillus cereus* group system, we compared the host ranges of phages isolated on one or four hosts and found that there was no difference in the breadth of host ranges of the isolated phages. Both narrow and broader host range phage were also equally likely to be isolated from either isolation procedure. While there are methods that reliably isolate broader host range phages, such as sequential host isolation, and there are other reasons to use multiple hosts during isolation, multiple hosts are not a consistent way to obtain broader host range phages.

## 1. Introduction

Bacteriophages, like all viruses, are constrained to only infect certain cells. These constraints are due to variation in phage and bacterial host molecules and biochemical pathways [1] and this property of only infecting certain species and strains describes the bacteriophage’s host range. While the host range is sometimes seen as primarily a function of the presence or absence of an appropriate receptor for the phage to attach to the host cell, defense mechanisms such as CRISPR and restriction enzymes can also affect host range by destroying phages as they infect the cell [2,3].

Host range is typically described as being narrow or broad but there are no accepted standards for either of these terms [4]. It is an important property for applications such as phage therapy, the use of bacteriophages in treatment of bacterial infections [5,6]. When phages are being isolated for applications such as phage therapy, phages with a wider host range are often seen as desirable because they can be used to treat more strains of a pathogenic species [7]. In situations where it is not practical to isolate the presumptive infecting pathogen for phage sensitivity testing, using phage with a wider host range increases the likelihood of a successful treatment. Similarly, broader host range phages, especially when mixed in phage cocktails, can be beneficial in creating standardized phage therapy agents that are meant to be used without diagnosing the infection to a particular host [7,8].

While all phages are limited in their host range, broader host range phages have been isolated and noted in the literature. In one early example, Lazarus and Gunnison [9] studied a phage isolated on *Pasteurella pestis* (now *Yersinia pestis*) that could broadly infect 63 of 74 strains of *P. pestis* and *Pasteurella pseudotuberculosis* (now named *Yersinia pseudotuberculosis* but now known to be a strain of *Y. pestis*). They also found that this phage could infect a limited number of *Salmonella* (3 of 42) and *Shigella* (6 of 37) strains. All of these species are related and classified within the order *Enterobacterales*.

Methods have been published for selection for broader host range phages. These include the simultaneous use of multiple hosts [10,11,12] or sequentially isolating collections of phages on a series of hosts [13]. Our lab has previously published data that supported the use of multiple hosts simultaneously to isolate broader host range phages, but the number of hosts and phages was small, so the results were suggestive at best [4]. In the work presented here, we directly compared the isolation of phages on single and multiple hosts. We chose to work with bacterial hosts from the *Bacillus cereus* group of bacteria as these represent a closely related group of bacteria but containing distinct species including *Bacillus mycoides, Bacillus thuringiensis, and Bacillus weihanstephanensis,* that were readily available [14]. The group includes some human and animal pathogens including *B. cereus* and *Bacillus anthracis* as well as non-pathogens (to humans) such as *B. thuringiensis* which produces a potent insecticidal protein Bt toxin. Since many members of the group are found in soil, we used multiple soil samples to isolate phages from, increasing the likelihood that many distinct phages could be found [15].

## 2. Materials and Methods

### 2.1. Bacteria and Growth Conditions

We obtained all the species and strains we used from the Bacillus Genetic Stock Center (BGSC, The Ohio State University, Columbus, OH, USA). Table 1 shows the 19 bacteria used in this study.

All bacteria were grown using nutrient broth and nutrient agar and incubated at 30 °C except for *B. cereus* 6A16 and *B. mycoides* 6A68 which grew more densely in broth cultures at room temperature (~18 °C) and *B. thuringiensis* 4A7 which grew best at 37 °C.

### 2.2. Isolation of Phages from Soil

Soil samples were collected from layers about 5–10 cm deep from farm fields or beneath grass in lawns. Most soil samples came from sites within 20 miles of Ashland, Ohio but a few were obtained from northern Illinois, southern Wisconsin, and western Washington state.

To isolate phages, we used a protocol modified from van Twest and Kropinski [16]. About 7–8 g of soil was transferred to a 50 mL conical tube. Approximately 10–15 mL of nutrient broth media was added to the tube to bring the total volume of soil plus broth to ~25 mL. For phage isolations with one or four host species, 1 mL of an overnight culture of each species was added to the tube. Tubes were incubated in a shaking incubator overnight at 30 °C unless otherwise indicated above. The next day, tubes were centrifuged for 10 min at 3000 rpm to pellet the remaining soil granules and bacteria. The cleared supernatant was then passed through a 0.45 μm syringe filter. The filtrate was then tested to detect the presence of phages.

### 2.3. Detection of Phages and Isolation of Pure Strains

Phages were detected using the agar overlay method of Adams [17] except that 0.6% soft agar was used instead of 0.7%. For this, various dilutions of the filtrate were mixed with bacteria in molten soft agar and poured onto the hard agar surface of a plate. After the soft agar had solidified, plates were incubated overnight at 30 °C unless otherwise indicated above. The next day they were examined for plaques.

To ensure that pure strains of phages were isolated, plaques were cored from the agar using a sterile Pasteur pipet and resuspended in 100 µL M9 salts buffer at either room temperature for several hours or overnight at 4 °C [17]. If multiple cores were taken from the same plate, plaques with different morphologies (clear vs. turbid, large vs. small, etc.) were collected to try to avoid isolating the same phage twice. No cores were combined in the same suspension. The suspensions were used to make dilutions to plate for a new passage. At least three rounds of passaging were done if the plaques had a consistent morphology in each passaging. If plaques with differing morphologies were seen, individual plaques were used to begin the isolation procedure again.

### 2.4. Preparation of Phage Stocks

Phage stocks were prepared using either the broth or plate lysate methods adapted from Carlson and Miller [18] except that chloroform lysis of cells was omitted as enveloped, filamentous, and some tailed phages can be inactivated by chloroform [19,20]. Phage stocks were initially prepared using the broth culture method. Stocks of phages that consistently had poor yields in broth were prepared using the plate lysate method.

### 2.5. Host Range Testing and Efficiency of Plating (EOP)

Phage host range was determined using two tests following the method outlined by Kutter [21]. The first test was a spot test with a high titer of phage stock. A fresh culture of bacteria was applied to a nutrient agar plate in molten top agar. After the top agar layer solidified, a 5 µL drop of phage stock (10^7^ pfu/mL or greater) was placed on the surface and allowed to absorb into the top agar. The plate was then incubated overnight at 30 °C. The next day, plates were examined to see if bacteria were killed by the phages. If no spot was seen, the bacteria was determined to not be within the phage’s host range and additional testing was not done with these phage–host combinations. If a clear or turbid spot was observed, for the second test, a serial dilution of phages was plated using the same drop method. Only bacteria that had bacteriophage plaques at some dilutions were noted as being within the phage’s host range. Spot testing was done once while plaque testing was done in triplicate.

For the phages listed below, efficiency of plating (EOP) was determined at the same time as testing for the plaque host range. EOP was calculated as the ratio of the apparent titer by plaque count on a host divided by the titer on the phage’s isolation host.

## 3. Results

### 3.1. Isolation of Phages

We isolated and analyzed 23 phages. Ten were isolated using only *B. cereus* 6A3 as the enrichment culture host and isolation host. Ten others were isolated either using a mixture of *B. cereus* 6A3*, B. mycoides* 6A11*, B. weihenstephanensis* 6A23*, and B. thuringiensis* 4A8 or a similar mix with *B. mycoides* 6A47 instead of strain 6A11. We were able to isolate phages on strains 6A3, 6A11, 6A47, and 4A8. No phages against strain 6A23 were found in either four-host isolation procedure using 6A23 as the isolation host (although phages isolated on other hosts can infect this strain). In addition, three phages were isolated using 6A3 from soil that was incubated overnight, by mistake, without any enrichment bacteria added. These phages may have been growing on bacteria endogenous to the soil or just washed off of soil particles.

### 3.2. Host Range Testing

We determined the host range of each phage using a collection of 19 *B. cereus* group bacteria (see Table 1). Bacteria were considered within a phage’s host range only if the phage could form plaques on that host after being diluted. This is described as the plaquing host range [1] and for most phages, is the same as the phage-productive host range. The results of this testing are summarized in Table 2.

Contrary to the expectation that using multiple hosts would tend to lead to isolation of broader host range phages, we found a breadth of host ranges in both the single-host and multiple-host isolated phages. For the single-host isolated phages, individual phages infected 2–14 hosts out of 19 hosts. For the multiple-host isolated phages, 1–15 hosts could be infected. The average number of hosts infected was 7.1 for single-host isolated phages and 6.4 for multiple-host isolated phages. We used a one-tail *t*-test to see if the difference was significant and it was not (*p* = 0.39).

While most phages either formed plaques after dilution or had no effect on the tested hosts, a few (AUBC1, AUBC12) were able to lyse hosts when the phages were at high concentration but did not produce plaques at any concentration. This can be described as the spotting host range of a phage but there are a number of reasons that this might occur that have nothing to do with phage infection such as the presence of endolysins from the phage stock production [4]. In addition, on two occasions, a single clear plaque was seen in a turbid high concentration spot. These were found to be host range mutant phages and did not indicate that a particular host was in the parent phage’s host range as they were not seen every time that phage–host combination was tested. The mutant phages were isolated and did have altered host ranges but were not further studied in this work.

The three phages isolated without an added enrichment host were able to infect four or five hosts. Given the small number of phages, it is not clear if this difference in breadth of host range from the single-host and multiple-host isolated phages is significant or not.

Similarly, the significance of one other observation is unclear. All three of the phages isolated using 6A3 as the host in the multiple-host isolations were all broader host range phages compared to those isolated using 6A11, 6A47, or 4A8; but the small number of phages makes any general conclusions difficult especially as phage 913-6A47-2, isolated on 6A47, has a nearly as broad host range.

One final conclusion can be drawn from Table 2, which is that most of the phages are distinct from each other in host range pattern, indicating that they are not independent isolates of the same phage. The exceptions are AUBC6 and AUBC11, which infect the same 14 hosts; AUBC7 and AUBC9 which infect the same two hosts; and the three phages isolated on *B. thuringiensis* 4A8, which only infect that strain of bacteria. Yet, if those potentially identical phages are removed from analysis, the overall results are the same. In this case, the average host range for the single-host isolated phages is 6.9 and for the four-host isolated phages it is 7.8. These revised host range average values are still not significantly different (*t*-test, *p* = 0.38)

### 3.3. Efficiency of Plating (EOP) Measurements

We chose six phages isolated using no added host (811-41), a single host (810-4A, AUBC12), or four hosts (528A, 528B, 528C) to measure the EOP for the bacteria within their phage-productive host ranges. The results are shown in Table 3. In this table, values of EOP were normalized against the bacterial strain that was used for the initial propagation of the phage when it was isolated. As with the overall host range results, there did not seem to be any general pattern with both the single-host and four-host isolated phages having a range of EOPs. Sometimes a phage grew more efficiently on the tested host (EOP > 1) and sometimes it grew less efficiently (EOP < 1).

## 4. Discussion

### 4.1. Methods for Finding Broader Host Range Phages

A number of methods have been proposed or developed to find broader host range phages. One well cited paper is that of Jensen and colleagues [10] who were studying and isolating phages against *Sphaerotilus natans, Escherichia coli*, and *Pseudomonas aeruginosa*, all of which are commonly found in biofilms at sewage treatment plants, within contaminated water, etc. [22]. They found that if they used any combination of two of the hosts, phages able to infect strains of both hosts were obtained. However, they also examined phages that were previously isolated on *S. natans* alone and found that nine of ten of those phages also infected both *S. natans* and *P. aeruginosa*. This suggests that broader host range phages can be found in environments with narrow host range phages rather than being selected for or evolving during enrichment culturing.

This mirrors what was found earlier by Green and Goldberg [23] who isolated 37 bacteriophages using a strain of *Streptomyces avermitilis* as the isolation host. When each was tested for host range on a bank of 11 species of *Streptomyces*, phages that infected 2 to all 11 species were identified.

Multiple other studies have isolated phages on mixtures of hosts although not with the stated purpose of isolating broad host range phages. These have used a variety of host systems as shown in Table 4. All found a breadth of numbers of hosts susceptible to infection by various phages. For example, Yildirum and colleagues [24] isolated 33 phages that they tested against 29 *Salmonella* strains and found between 10% to 62% of the host strains were susceptible to infection. Similarly, Betz and Anderson [25] isolated 12 phages infecting *Clostridium sporogenes*. These phages were tested against 25 strains of *C. sporogenes* and 48% to 80% of the strains were sensitive to some phages. Oliveira and colleagues [11] isolated five *E. coli* phages and tested them against a bank of 148 host strains, mostly field isolates and 24–48% of those bacteria were susceptible to phages. With our results, these earlier works support the conclusion that broader host ranges are not being selected for but if sufficient phages are isolated, some of these will have broader host ranges by chance. This is consistent with the observation that bacteriophages in natural systems may be specialists (narrow host range) whereas others may be generalists (broader host range) in terms of the hosts they can reproduce on [26,27].

One caveat to comparing our results to the studies cited in Table 4 is that most of the studies cited relied on spot testing for host range determination rather than plaque testing. Only two of the studies [31,33] seem to have tested for plaque production as they note EOP values for the phages tested. As we [4] and others [37] have noted, spot testing tends to overestimate host range compared to plaque testing. Of the two studies that did determine host range, like our results, they found varying EOPs on the hosts within a phage’s host range.

Isolation methods that consistently yield broader host range phages have been published. Yu and colleagues [13] were able to isolate broader host range phages by sequentially exposing a pool of phages from environmental samples to multiple hosts in either of two procedures. Cultures of the phages in the pool that could grow on the first host were then grown on the next host. Only the phages that grew on the second host were used to grow a pool on the third host and so on. In the final step, they isolated pure phage cultures through plaque isolation. They demonstrated these protocols isolated phages that could infect multiple strains of *E. coli* and multiple species of *Pseudomonas* but the procedure should work for other combinations of bacteria.

A different approach is to expand the host range of previously isolated bacteriophages. This has been shown in the context of developing phages that can overcome evolved host resistance to infection using a cocktail of phages [38,39]. This procedure, a modification of the Appelmans protocol [40], adds a cocktail of phages to a mixture of resistant and sensitive bacteria. After incubation, lysates are collected, pooled, and added to a fresh mix of hosts. After 30 cycles, phages can be isolated that are able to infect the resistant hosts. Genome analysis shows that these phages are recombinants of some of the original cocktail phages [38]. While this is not a novel phage isolation method, it may in principle be used to develop broader host range phages using a mix of sensitive and insensitive hosts although it may be that there will be a change of host range within the hosts used rather than an overall broadening of host range.

A third approach is to use genetic engineering techniques on an isolated phage to alter its host range by targeted changes to the receptor binding proteins—usually, tail fibers or tail spikes. A number of studies have shown that the host range of a phage could be altered by replacing the receptor binding region with its equivalent from another phage [41,42,43]. In these cases, the host range was not necessarily expanded, just changed to that of the receptor binding protein donor. A different approach was demonstrated by Pouillot and colleagues [44]. They cloned genes 37 and 38 of bacteriophage T4 which encode the end of the tail fiber including the critical binding regions (gene 37) and the accessory protein for tail fiber assembly (gene 38). They then used a site directed mutagenesis technique to create a library of plasmids with alterations in the key sequences for receptor binding. Once these altered genes were transferred back to phages by homologous recombination, they obtained a library of phages including ones that could infect *Yersinia ruckeri* and others that infected *P. aeruginosa*. Again, however, each individual phage may have a narrow or broader host range but the library as a whole contained phages able to infect many different bacteria.

### 4.2. Should Multiple Hosts Be Used to Isolate Bacteriophages?

Manuals on phage procedures differ on whether to use a single host [16,17,45] or multiple hosts [18,46]. The two protocols recommending multiple hosts explicitly mention that this should yield broader host range phages. If, as indicated by our results, this is not a consistent outcome, are there other benefits to using multiple hosts in phage isolation? It seems likely that having multiple hosts will increase the chance of finding at least some phages in a particular environmental sample, although we are not aware of any tests of this hypothesis. There is the caveat that the multiple hosts should be tested for antagonistic interactions so that all can grow in the same culture, but if these antagonisms are not present, using multiple hosts should be equivalent to doing multiple isolation procedures with a different host in procedure. To maximize the chances of isolating phages, the putative phage culture should be tested on each of the enrichment hosts individually as the newly isolated phages will likely have only partially overlapping host ranges as indicated by our results and the studies cited in Table 4. Host range testing would be useful as a simple test for the unique identity of each newly isolated phage. Any phages with identical host ranges would need additional screening to ensure that the same strain of phage isn’t isolated multiple times on each host within its host range.

### 4.3. Limitations and Future Studies

While we are confident of our overall conclusions, there are some limitations to this study. As we have suggested elsewhere [7], host range characterization should use as large a group of hosts as possible. Nineteen hosts is a modest number in this regard. There is also a chance that the results were influenced by a lack of diversity among the various hosts. Genomic sequences are only available for a few of these strains specifically, making it difficult to judge this without a significant effort in genome sequencing. However, we do note that our results suggest diversity in the hosts. Host range testing, when organized by host rather than phage (that is, reading rows rather than columns in Table 2) is phage typing, the use of bacteriophage susceptibility to differentiate bacteria [47]. By this standard, the 19 hosts we used are distinct. None have an identical pattern of susceptibility to these 23 phages.

The way to resolve these questions, as well as the ambiguities described at the end of Section 3.2, is to expand these results with more phages and more hosts, especially the latter. These newly isolated phages are also not yet characterized biologically or genome sequenced. This is not essential for the conclusion on isolating broader host range phages but every newly isolated phage has the potential to provide interesting novel genes, as well as expanding our understanding of phage biology.

## 5. Conclusions

Our results indicate that using multiple hosts does not select for broader host range phages. This does not mean that using multiple hosts is deleterious in some way. Numerous studies have used multiple hosts to successfully isolate phages. If the goal of phage isolation, though, is to specifically obtain broader host range phages, then the number of isolation hosts is not important. One can either use a more specific broader host range protocol described above or one can simply determine the host range of all the newly isolated phages. Some will have a narrow host range and some will be broader in host range. Whether single or multiple hosts are used, host range testing is an essential part of phage characterization.

## Figures and Tables

**Table 1 viruses-15-00518-t001:** Bacterial species and strains used in this study.

Species	BGSC Number	Original Designations
*B. cereus*	6A1	Strain T
	6A2	T-HT
	6A3	NRRL B-569
	6A15	ATCC 10987, NRS 248
	6A16	
	6A17	ATCC 13472
	6A100	RS438, CDC2000032805
*B. mycoides*	6A11	95/1883
	6A13	NRS 306
	6A47	WSBC10277
	6A68	STR4
*B. pseudomycoides*	6A49	WSBC10360
*B. thuringiensis* serovar *thuringiensis*	4A1	NRRL B-4039
	4A7	Bt1
	4A8	Bt131
	4A9	Bt1627
*B. weihenstephanensis*	6A21	10204
	6A22	10206
	6A23	10396

**Table 2 viruses-15-00518-t002:** Host ranges of isolated phages on various *Bacillus* species and strains as determined by plaque formation and spot testing.

	Virus	AUBC1	AUBC2	811-4A	AUBC4	AUBC5	AUBC6	AUBC7	AUBC8	AUBC9	AUBC10	AUBC11	810-4A	AUBC12	528A	913-6A3-1	913-6A3-2	528B	528C	913-6A47-1	913-6A47-2	913-4A8-1	913-4A8-2	913-4A8-3
	Number of enrichment hosts	0	0	0	1	1	1	1	1	1	1	1	1	1	4	4	4	4	4	4	4	4	4	4
	Propagation host	*B. cereus* 6A3			*B. cereus* 6A3										*B. cereus* 6A3			*B. mycoides* 6A11		*B. mycoides* 6A47		*B. thuringiensis* 4A8		
*B. cereus*	6A1	T	T	N	N	N	P	N	P	N	N	P	T	C	P	P	P	N	N	N	P	N	N	N
	6A2	T	N	N	P	P	P	N	P	N	N	P	T	T	P	P	P	N	N	N	P	N	N	N
	6A3	P	P	P	P	P	P	P	P	P	P	P	P	P	P	P	P	P	N	N	P	N	N	N
	6A15	N	N	N	N	N	N	N	N	N	N	N	N	T	N	N	N	N	N	N	N	N	N	N
	6A16	N	N	N	N	N	P	N	N	N	P	P	N	N	N	N	N	N	N	N	N	N	N	N
	6A17	P	P	N	P	P	P	N	P	N	P	P	P	P	P	P	P	P	P	N	P	N	N	N
	6A100	T	N	N	N	N	N	N	N	N	N	N	P	T	P	P	P	N	N	N	T	N	N	N
*B. mycoides*	6A11	N	N	N	N	P	P	N	N	N	N	P	P	N	P	P	P	P	P	N	N	N	N	N
	6A13	N	N	N	N	P	P	N	N	N	N	P	P	N	P	P	T	P	N	N	N	N	N	N
	6A47	P	P	P	N	P	P	N	P	N	N	P	P	T	P	P	P	N	N	P	P	N	N	N
	6A68	N	N	P	N	N	N	N	N	N	N	N	P	T	N	P	P	N	N	N	P	N	N	N
*B. pseudomycoides*	6A49	N	N	N	N	P	N	N	N	N	N	N	N	N	P	P	N	N	N	N	N	N	N	N
*B. thuringiensis*	4A1	N	N	N	N	P	P	N	N	N	N	P	P	T	P	N	N	N	N	N	N	N	N	N
	4A7	P	N	N	N	P	P	N	N	N	N	P	P	T	P	P	P	P	N	N	P	N	N	N
	4A8	T	T	N	N	P	P	N	N	N	N	P	P	T	N	N	P	N	N	N	P	P	P	P
	4A9	N	N	N	N	P	P	N	N	N	N	P	P	N	P	P	P	P	N	N	N	N	N	N
*B. weihensteph-anensis*	6A21	N	N	N	N	N	N	N	N	N	N	N	T	N	P	T	N	N	N	N	P	N	N	N
	6A22	C	T	P	N	N	P	P	N	P	N	P	P	C	P	P	P	N	N	N	N	N	N	N
	6A23	P	P	N	N	P	P	N	P	N	N	P	P	N	P	P	P	N	N	N	P	N	N	N
	Sum of hosts in host range	5	4	4	3	12	14	2	6	2	3	14	13	2	15	14	13	6	2	1	10	1	1	1

Host range key: P: plaques with dilution (green); N: no growth (red); C: clear zone of lysis at high concentration only (orange); T: turbid zone of lysis at high concentration only (orange).

**Table 3 viruses-15-00518-t003:** Efficiency of plating of selected phages on various hosts.

		0	1	1	4	4	4	Number of Enrichment Hosts
		*B. cereus* 6A3				*B. mycoides* 6A11		Propagation host
Bacterial species and strain		811-4A	810-4A	AUBC12	528A	528B	528C	Phage
*B. cereus*	6A1	–	–	–	3.85	–	–	
	6A2	–	–	–	2.31	–	–	
	6A3	1.00	1.00	1.00	1.00	0.03	–	
	6A15	–	–	–	–	–	–	
	6A16	–	–	–	–	–	–	
	6A17	–	4.55	–	1.54	0.01	0.50	
	6A100	–	9.1	1.20	0.04	–	–	
*B. mycoides*	6A11	–	0.70	–	3.85	1.00	1.00	
	6A13	–	0.92	–	1.54	0.002	–	
	6A47	1.60	1.04	–	ND	–	–	
	6A68	1.68	0.50	–	–	–	–	
*B. pseudomycoides*	6A49	–	–	–	0.02	–	–	
*B. thuringiensis*	4A1	–	0.52	–	3.85	–	–	
	4A7	–	0.91	–	2.31	0.05	–	
	4A8	–	0.54	–	–	–	–	
	4A9	–	0.68	–	1.15	0.15	–	
*B. weihenstephanensis*	6A21	–	–	–	0.92	–	–	
	6A22	ND	1.50	–	1.23	–	–	
	6A23	–	5.35	–	ND	–	–	

ND: not determined.

**Table 4 viruses-15-00518-t004:** Results of phage isolation using multiple hosts from previously published studies.

Phage Host Target Species	Number of Strains Used for Isolation	Number of Phages Isolated	Number of Strains Tested for Phage Sensitivity *	Results of Host Range Testing	Reference
*Escherichia coli*	8 *E. coli* strains	5 phages isolated	148 host strains tested	24% to 48%, of hosts were sensitive to phage infection	[11]
*Salmonella typhimurium* and/or *Salmonella enteritidis*	25 *S. enteritidis* strains	11 phages isolated	31 host strains tested	21 (68%) to 30 (97%) of hosts were sensitive to phage infection	[12]
	Mix of 12 *S. typhimurium* and 10 *S. enteritidis* strains	33 phages for *Typhimurium* and 56 phages for *Enteritidis* isolated	29 hosts strains tested	Phages killed 3 (10%)–18 (62%) of 36 *Typhimurium* and 2 (6%)–15 (42%) of 36 *Enteritidis* strains. 20 of the phages also killed *E. coli*	[24]
	Mix of 7 *Salmonella* serotypes and 1 *E.coli* strain	4 phages isolated	234 host strains tested	172 (74%) to 214 (91%) strains sensitive to phages	[28]
*Campylobacter coli*	12 *C. coli* strains	43 phages isolated	15 host strains test	20–93% of strains sensitive to phages	[29]
*Pseudomonas aeruginosa*	4 *P. aeruginosa* strains	17 phages isolated	35 host strains tested	17–89% of strains sensitive to phages	[30]
*Pseudomonas fluorescens*	3 *P. fluorescens* strains	7 phages isolated	23 host strains tested	1 (4%) to 3 (13%) of hosts sensitive to phage	[31]
*Staphylococcus epidermidis*	Number not stated	5 phages isolated but only 1 phage characterized	67 host strains tested (41 *S. epidermidis),* 22 other *Staphylococcus* species, 3 *Bacillus* and one *Listeria* species	41/41 *S. epidermidis* but only 2 of remaining species	[32]
*Yersinia enterocolitica and Yersinia pseudotuberculosis*	6 *Yersinia* strains	Many phages isolated, 18 characterized for host range	94 host strains tested	1 (1%)–8 (9%) infected ^†^	[33]
*Proteus mirabilis*	13 *P. mirabilis* strains	2 phages isolated	43 host strains tested	37% and 60% infected	[34]
*Morganella morganii*	27 *M. morganii* strains	2 phages isolated	27 *M. morganii* strains + 6 others	19 (70%) and 15 (56%) of *M. morganii* test strains infected, no others	[35]
*Clostridium sporogenes*	Not explicitly stated, “isolated by the mixed-culture technique”	12 phages isolated	25 host strains tested	48–80% of strains were sensitive	[25]

* Unless otherwise indicated, all the host range testing strains (column 4) were strains of the phage target species (column 1). ^†^ In a later study [36], three of these phages were further tested and were found to infect a broader range of *Yersinia* species as well as some *E. coli* strains.

## Data Availability

Not applicable.

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
