# Peer review of "More’s the Same—Multiple Hosts Do Not Select for Broader Host Range Phages"

_viruses, 2023, doi:10.3390/v15020518_

Round 1

Reviewer 1 Report

The crucial part of the manuscript 2.5. Host range testing and efficiency of plating (EOP) needs references to the methods used. 

All methods section needs references to the methods used.

Author Response

The crucial part of the manuscript 2.5. Host range testing and efficiency of plating (EOP) needs references to the methods used. 

All methods section needs references to the methods used.

Thank you for pointing this out.  Additional references have been added to sections 2.2, 2.3 and 2.5.

Reviewer 2 Report

In this study, authors isolated phages against Bacillus species and used one host and multiple hosts for isolation. They compared the results of host range and EOP for the isolated phages according to the number of hosts used in isolation.

Comments:

1-The title of table 1 was repeated exactly in the method section (lines 63-65), and it is too long. I think it should be more abbreviated.

2-The two methods of spot test and EOP should be separated into two paragraphs. The authors explained the spot test in detail, but the EOP test was not the same.

3- The authors mentioned that they did spot tests and plaquing host range, why did they use the two methods?

I think the plaquing host range is enough to ensure phage productivity.

4-The authors stated that they isolated 23 novel phages. Why are these phages considered novel?

They should highlight the reasons for the novelty.

5- In line 123, the authors mentioned that no phages against strain 6A23 were found in either four-host isolation procedure, while in table 2, this strain was infected with some of the isolated phages like 913-6A31, 913-6A32, and others and these phages showed plaques when spotted on this strain. I do not understand this point. Can the authors explain this sentence?

6- In the host range experiment, authors used only 19 strains from the same species. Do the authors think that this number is enough compared to the other studies they mentioned in table 3?

Also, they should test the host range of these phages against other bacteria from other genera. Did the authors do this?

7- The discussion is well explained, but it did not cover all results obtained. Please, discuss the results of the EOP test and link it with prior publication.

8- For the latter, please highlight potential areas for future studies.

9- Further characterization of isolated phages should be included in the manuscript. At least, the TEM micrographs of phages and if possible, the one-step growth curves.

10- Phage genomes or at least some of them needed to be sequenced to ensure novelty and included in the manuscript.

 11- The authors concluded that multiple hosts in isolation are still important, and it is preferable, so what is this study bringing new to the phage research?

Author Response

1-The title of table 1 was repeated exactly in the method section (lines 63-65), and it is too long. I think it should be more abbreviated.

Thank you for pointing this out.  I’ve removed the source sentence from the table legend.

2-The two methods of spot test and EOP should be separated into two paragraphs. The authors explained the spot test in detail, but the EOP test was not the same.

Thank you for this comment.  EOP is calculated using the results from the plaque testing plates so there is not a full separate method.  I have split the paragraph as suggested and clarified the source of the plaque counts.

3- The authors mentioned that they did spot tests and plaquing host range, why did they use the two methods?

I think the plaquing host range is enough to ensure phage productivity.

You are correct that plaquing host range is essential to show phage productivity.  The initial spot tests were done to rapidly identify all phage-host combinations that produced no spots and therefore could be eliminated from the more time and plate consuming plaque tests.  I have added additional text to make this clearer (lines 113-116).

4-The authors stated that they isolated 23 novel phages. Why are these phages considered novel?

They should highlight the reasons for the novelty.

You are correct that we would need additional data to support that term. The word novel has been deleted in line 124 and in the section 4.2 title

5- In line 123, the authors mentioned that no phages against strain 6A23 were found in either four-host isolation procedure, while in table 2, this strain was infected with some of the isolated phages like 913-6A31, 913-6A32, and others and these phages showed plaques when spotted on this strain. I do not understand this point. Can the authors explain this sentence?

Thank you for pointing out this ambiguous sentence.  We meant to say that no phages were isolated using 6A23 as the isolation host, not that no phages could infect it.  This has been clarified in lines 129-130.

6- In the host range experiment, authors used only 19 strains from the same species. Do the authors think that this number is enough compared to the other studies they mentioned in table 3?

Also, they should test the host range of these phages against other bacteria from other genera. Did the authors do this?

A point of clarification.  We did not use 19 strains of the same species but rather used multiple strains of five separate species for a total of 19 as indicated in table 1.  I have made this more explicit in line 63 and the table 1 legend as well as replacing the word strain in several other places for clarity.  Thank you for drawing my attention to the need to be clearer about this.

To your main point, ideally yes, we would have liked to test more bacteria including isolating other B. cereus group strains from soil but this work as well as testing on other genera.  Practical considerations limited our work to these bacterial species and strains.  This work was started in 2019, interrupted in 2020 by our university shutting down due to covid and then on limited access for the rest of that year when we reopened.  As well, all the students working on this project were undergraduates who have now graduated (we have no graduate program at AU).  Additional work would require recruiting new students and training them.

Compared to the studies listed in table 4 and in the text, this work is at the low end but not the lowest. Two studies used fewer hosts - Campylobacter coli phages (15 hosts, reference 29) and studies by Green and Goldberg (lines 203-206) (11 hosts, reference 23).  Additionally, four other studies in table 4 used less than 30 hosts so we feel the 19 hosts we used are adequate if not ideal.

I have added a section (4.3) on limitations to this study including that more hosts would strengthen the results.

7- The discussion is well explained, but it did not cover all results obtained. Please, discuss the results of the EOP test and link it with prior publication.

Only two of the studies cited in Table 4 reported EOP numerical values making comparisons challenging.  We have noted this in the added text in lines 225-231 which also notes that many previous studies used spot testing rather than plaque testing.

8- For the latter, please highlight potential areas for future studies.

Future studies are now mentioned at the end of the new section 4.3

9- Further characterization of isolated phages should be included in the manuscript. At least, the TEM micrographs of phages and if possible, the one-step growth curves.

While these are likely to be interesting and something we may work on in future studies, I do not feel they are essential for this study.  This paper is focused on testing a claim about the host range of newly isolated phages.  Other properties of the phages are not going to affect that conclusion. These sorts of additional research are noted in the last paragraph of the Limitations and future studies section.  As I noted in responding to question 6, additional work will require recruiting and training new students and could not be completed until at least next summer or beyond.

10- Phage genomes or at least some of them needed to be sequenced to ensure novelty and included in the manuscript.

As noted in our response to the previous question, I feel such information is interesting but not essential the conclusions about isolating phages and host range.

 11- The authors concluded that multiple hosts in isolation are still important, and it is preferable, so what is this study bringing new to the phage research?

I appreciate the question.  This is, of course, one of the challenges in reporting negative results.  We feel that it is important that, even if phage researchers continue to use multiple hosts, they know what they should and should not expect from the phage produced.  Especially that they shouldn’t rely on the multiple host use to necessarily produce broader host range phages.  We have re-written the conclusions to better state this.

Reviewer 3 Report

The manuscript addresses the issue of isolating broad host range phages (BHRP) in the presence of one or multiple hosts. Despite the apparent success of isolating BHRPs in the presence of various hosts reported previously, a systematic study comparing BHRPs selected in the presence of single versus multiple hosts has not been done. The author's experimental design using Bacillus species as a model wants to distinguish between the broad host range phages outcomes in the presence of one or four species.

The results indicate no significant difference in the host range of phage isolates using a single host or a mix of hosts. 

Some possible limitations of the study are:

1.     Bacillus genetic diversity could bias the estimation of the host range. If the genetic distance between the Bacillus species is tight, it could be unsurprising to see similar host ranges for the phages in both methods. May the author comment on the genetic or phylogenetic distance between species and strains used? Perhaps adding this information would provide a genetic context to the infection host range.

2.     Although the experiment is well designed to distinguish the influence of one or more hosts on the isolation of BHRPs, the number of experiments using four hosts is still low to conclude. For instance, four BHRPs were obtained when using strain 6A3 alone. In contrast, in a mix of 6A3 with three additional hosts, all the isolated phages for this strain are BHRP. It also seems that in the presence of other hosts, there is a preference to select BHR phage for 6A3. It could be possible to test the consistency of this finding by performing more assays. How did the authors evaluate this result? 

3.     Minor comments:

1.     There is no information on the number of replicates of the spot-test experiments.

2.     Tables 2 and 3 could be improved for easy reading, formatting the size of rows and fonts.

Author Response

Some possible limitations of the study are:

  1. Bacillus genetic diversity could bias the estimation of the host range. If the genetic distance between the Bacillus species is tight, it could be unsurprising to see similar host ranges for the phages in both methods. May the author comment on the genetic or phylogenetic distance between species and strains used? Perhaps adding this information would provide a genetic context to the infection host range.

 This is an interesting idea.  Unfortunately, sequence information for many of the hosts does not appear to be available.  However, we do have some data from this study that the bacteria are distinct and I have added this to the new Limitations and future directions section (4.3) (lines 286-296).

  1. Although the experiment is well designed to distinguish the influence of one or more hosts on the isolation of BHRPs, the number of experiments using four hosts is still low to conclude. For instance, four BHRPs were obtained when using strain 6A3 alone. In contrast, in a mix of 6A3 with three additional hosts, all the isolated phages for this strain are BHRP. It also seems that in the presence of other hosts, there is a preference to select BHR phage for 6A3. It could be possible to test the consistency of this finding by performing more assays. How did the authors evaluate this result? 

Thank you for this observation.  I have added text to the results section noting this (lines 165-169).  As you say, more studies would be needed to see if this is more than coincidence.  I have also noted this in the new Limitations and future directions section (4.3).

  1. Minor comments:

  1. There is no information on the number of replicates of the spot-test experiments.

Thank you for pointing this out.  The information has been added to line 118.

  1. Tables 2 and 3 could be improved for easy reading, formatting the size of rows and fonts.

I completely agree.  The tables in the review manuscript are not formatted as they are in the submitted manuscript and I have already confirmed with the journal staff that it will be possible to reformat them during the proofing process, presuming that the article is accepted

Reviewer 4 Report

This manuscript is is clearly written and fluent to read. The experimental design is appropriate. The conclusions are sound. I only request one correction on line 230 (...gene 37 and gene 38 (instead 38 genes...).  Would be nice to have further characterization of the phages (such as their genomes) but I trust this will be performed in future work.

Author Response

This manuscript is is clearly written and fluent to read. The experimental design is appropriate. The conclusions are sound. I only request one correction on line 230 (...gene 37 and gene 38 (instead 38 genes...).  Would be nice to have further characterization of the phages (such as their genomes) but I trust this will be performed in future work.

Thank you for the complimentary review and for catching that error.  I have fixed it in line 258.  You are correct that we hope to better characterize some of the phages in the future.  As Ashland University is an undergraduate only institution and all four student authors have graduated, additional work will require recruiting and training new students so that work will not happen quickly unfortunately. 

Round 2

Reviewer 1 Report

revisions are made

Author Response

Thank you again for your review.

Reviewer 2 Report

Dear authors,

I highly appreciate your efforts and the time you spent doing this work, but I think the work still needs more experiments to complete. I understand the situation and difficulties during the last few years, but these problems should not affect the research quality. 

The number of hosts tested in this study is small and can not be used to draw a conclusion. 

Further characterization of bacterial hosts and phages must be done to enhance the quality of the manuscript and support your claims. 

phage host range is not enough to know if the isolated phages are similar or different. 

The authors used only B.cereus to draw the conclusion, but they did not test other bacteria species, and most studies refer to the importance of using more hosts in the isolation of phages with a broad host range. 

In my point of view, I recommend authors widen this work to include more different hosts to make a general conclusion about is no big difference when using one or more hosts if we want to isolate broad host phages.  

Author Response

Thank you for your comments.  I don't disagree with most of them and have acknowledged them in the Limitations.  I think the only disagreement is that I feel the numbers of phages and hosts are adequate to draw a conclusion, especially in light of our reappraisal of other studies.  But I do hope to expand this work with additional species in the future.